# Susceptibility to Degradation in Soil of Branched Polyesterurethane Blends with Polylactide and Starch

**DOI:** 10.3390/polym14102086

**Published:** 2022-05-20

**Authors:** Joanna Brzeska, Grzegorz Jasik, Wanda Sikorska, Barbara Mendrek, Jakub Karczewski, Marek Kowalczuk, Maria Rutkowska

**Affiliations:** 1Department of Industrial Product Quality and Chemistry, Gdynia Maritime University, 83 Morska Street, 81-225 Gdynia, Poland; m.rutkowska@wpit.umg.edu.pl; 2Non-Food and Packaging Laboratory, J.S. Hamilton Poland Sp. z o.o., Chwaszczynska Street 180, 81-571 Gdynia, Poland; gjasik@jsh.com.pl; 3Centre of Polymer and Carbon Materials, Polish Academy of Sciences, 34 M. Curie-Sklodowska Street, 41-819 Zabrze, Poland; wsikorska@cmpw-pan.edu.pl (W.S.); bmendrek@cmpw-pan.edu.pl (B.M.); 4Advanced Materials Center, Gdansk University of Technology, Narutowicza Street 11–12, 80-223 Gdansk, Poland; jakub.karczewski@pg.edu.pl; 5Faculty of Applied Physics and Mathematics, Institute of Nanotechnology and Materials Engineering, Gdansk University of Technology, Narutowicza Street 11-12, 80-223 Gdansk, Poland

**Keywords:** polyurethanes, poly([R,S)-3-hydroxybutyrate), poly([D,L]-lactide), starch, (bio)degradability, soil

## Abstract

A very important method of reducing the amount of polymer waste in the environment is the introduction to the market of polymers susceptible to degradation under the influence of environmental factors. This paper presents the results of testing the susceptibility to degradation in soil of branched polyesterurethane (PUR) based on poly([R,S]-3-hydroxybutyrate) (R,S-PHB), modified with poly([D,L]-lactide) (PLA) and starch (St). Weight losses of samples and changes in surface morphology (SEM, OM and contact angle system) with simultaneously only slight changes in molecular weight (GPC), chemical structure (FTIR and ^1^HNMR) and thermal properties (DSC) indicate that these materials are subject to enzymatic degradation caused by the presence of microorganisms in the soil. Chemical modification of branched polyesterurethanes with R,S-PHB and their physical blending with small amounts of PLA and St resulted in a slow but progressive degradation of the samples.

## 1. Introduction

Managing the environment and its condition requires taking steps to reduce the quantity and to change the quality of the generated waste. One of the most important methods of reducing the amount of polymer waste is the introduction to the market of products made of biodegradable polymers, which will decompose after the end of their service life under the influence of environmental factors. Polyurethanes (PURs) are such polymers, widely used in many industries. Currently, in Europe, the same amount of PURs (in various forms) is produced as the popular polyethylene terephthalate (e.g., as PET bottles) [1]. At the same time, popularity of PURs causes a huge amount of generated waste. This waste still often ends up in municipal or illegal landfills. Intensive research is being carried out to develop methods to reduce the amount of PUR waste in the environment. This research ranges from glycolysis leading to polyols [2], through the use of synthetic degradable substrates [3,4], to physical and chemical modification with natural substrates [5,6,7]. Of these methods, it seems clear that only plastics with on-demand degradation will be optional for products that potentially end up in the open environment [8].

Polymer biodegradation is the consequence of a number of physical, chemical and biological processes related to the destruction of polymers [9]. The final process is the action of enzymes secreted by microorganisms (bacteria and microscopic fungi) initially depositing on the surface of the sample.

As a result of the degradation, the polymer chains are shortened (the molecular weight of the polymer is reduced) and their fragments are eliminated, and consequently the weight of the polymer samples is reduced. The enzymatic degradation process itself takes place on the surface of the samples, in contrast to the chemical hydrolysis which runs throughout the sample bulk. Under favorable conditions for their development (in the presence of oxygen, moisture, microelements, at the appropriate temperature and pH), microorganisms secrete enzymes that initiate the depolymerization process, leading to the final decomposition of the polymer into monomers or to the formation of other low-molecular compounds, which are then absorbed by the microorganisms as nutrient material [10]. Several strains of bacteria and fungi capable of biodegradation of PURs have already been isolated [10,11,12,13]. However, the presence of these specific strains in ordinary garden or landfill soil is not obvious. PUR waste is more likely to end up in more or less enriched anthropological (garden, landfill) or natural soil.

The synthesis of PUR from compounds containing ester groups as well as its physical modification with polyester gives hope for obtaining a material susceptible to enzymatic hydrolysis. Among all polyesters, poly(d,l-lactide) (PLA), poly(β-hydroxybutyrate) (PHB) and poly(ε-caprolactone) (PCL) are considered to be biodegradable thermoplastic aliphatic polyesters [14]. In addition to crystalline PHB, its synthetic counterpart, amorphous, atactic poly([R,S]-3-hydroxybutyrate) (R,S-PHB), is also used to modify PURs [3]. R,S-PHB is obtained through anionic ring-opening polymerization (ROP) of β-substituted β-lactones [15]. PLA and R,S-PHB degradation products are non-toxic and can finally be biodegraded into carbon dioxide and water. Another broad group of compounds used to modify synthetic polymers, including PURs, is carbohydrates. Among them, starch is of increasing importance due to its biodegradability, wide availability and cheapness. As indicated by Tai et al., by modifying PUR with starch in various ways, it is possible to obtain materials susceptible to degradation, also in soil [16].

The intention of this work was to investigate whether these polyurethanes, under the influence of biopolymer modification, will degrade faster after the end of their service life. Unfortunately, in Poland we have a big problem with polyurethane waste being deposited in wild and municipal landfills. The conditions in these places are different; they cannot be averaged to determine if they are optimal for the biodegradation of these materials. We will also never have a guarantee that the environment will contain just microorganisms capable of assimilating such chemical compounds. Therefore, it seemed important to us to see whether in ordinary soil, where such polymer waste may happen by chance, polyurethane made of degradable substrates and additionally modified with biopolymers will actually biodegrade over time.

Thus, the main reason for the research described in this publication is to investigate the susceptibility of branched PUR materials modified with polyesters, R,S-PHB and PLA, and starch for degradation in anthropological soil. This paper is a continuation of our previous research on determining the susceptibility of these materials to degradation in a hydrolytic and oxidizing environment [17]. Degradability was determined using attenuated total reflectance Fourier transform infrared spectroscopy (ATR FTIR), proton nuclear magnetic resonance spectroscopy (^1^H NMR), gel permeation chromatography (GPC-MALLS), differential scanning calorimetry (DSC), scanning electron microscope (SEM), optical microscopy (OM) and a goniometer (static water contact system).

## 2. Materials and Methods

### 2.1. Materials

Poly([R,S]-3-hydroxybutyrate) diol (R,S-PHB) was synthetized via polymerization of β-butyrolactone (Sigma-Aldrich, St. Louis, MO, USA), using anionic ring-opening polymerization, initiated by 3-hydroxybutyric acid sodium salt/18-crown-6 complex (Sigma-Aldrich, St. Louis, MO, USA), at room temperature and terminated with 2-iodoethanolor 2-bromoethanol (Sigma-Aldrich, St. Louis, MO, USA) [18].

Two branched polyesterurethane films differing in the amount of R,S-PHB (M_n_ 1900) in the structure of soft segments and synthetized by the two-step method in the presence of Tin(II) octanoate (OSn, Alfa Aesar, Karlsruhe, Germany) and N,N’-dimethylformamide (DMF, POCH, Gliwice, Poland) [19] were tested. They contained 10 or 20 wt.% R,S-PHB, 5 wt.% poly(ε-caprolactone) triol (PCL_triol_) (M_n_ 900, Sigma-Aldrich, St. Louis, MO, USA) and 85 or 75 wt.% poly(ε-caprolactone) diol (PCL_diol_) (M_n_1900, Sigma-Aldrich, St. Louis, MO, USA), respectively, in the soft segments (Table 1). Hard polyurethane segments were synthesized from dicyclohexyl diisocyanate (H_12_MDI) (Sigma-Aldrich, St. Louis, MO, USA) and 1,4-butanediol (1,4-BD) (Sigma-Aldrich, Steinheim, Germany). A simplified scheme of the synthesis of polyesterurethanes and their blends is shown in Figure 1, and a scheme of the chemical structure of polyesterurethanes is shown in Figure 2. Blends of these polyurethanes with poly(dl-lactide) (PLA) (M_n_18,000–28,000, Sigma-Aldrich, Steinheim, Germany) and, in the case of polyurethane with a higher R,S-PHB content, also with starch (St) (Hengshui Fuqiao Starch Co., Hengshui, China) were prepared. A schematic representation of the polyesterurethane chains and biopolymer inclusions is shown in Figure 3.

### 2.2. Methods

The horticultural soil used for the study was currently not being cultivated, and therefore was not fertilized with natural or synthetic fertilizers. It was taken from a garden in the northern part of Poland (GPS coordinates: 54.528368455536864, 18.289577253812595), with a warm, temperate transitional climate. The soil parameters (conductivity, density and elemental composition) before and after the addition of vermicompost are given in Appendix A. The anthropogenic soil pH (7.27) was measured using a WTW pH meter equipped with a combined electrode operating in the pH range of 0–14 by SI Analytics according to BN-75 9180-03. The sieve analysis of the soil was performed on a shaker using sieves with a mesh diameter of 2.0; 1.0; 0.5; 0.2; 0.01; <0.1 mm. The soil was classified into the granulometric group—sand and the granulometric subgroup—loose sand (Appendix A) [20]. Appendix A shows the total number of bacterial and fungal colonies, and also the number of some soil-dwelling microorganisms (including those known as PUR-degrading—*Bacillus* spp. [13]). As checked, vermicompost introduced only a negligible amount of these microorganisms into the soil. Its task was to maintain appropriate conditions for the survival of microorganisms during the study. During such isolation, when the soil samples were placed in small containers in the climatic chamber, without the possibility of collecting nutrients from the outside, the proliferation of microorganisms would be difficult or even impossible. Moreover, sterilizing the soil and then inoculating only with microorganisms that degrade PUR materials would not reflect the actual conditions in the natural environment. The intention of the study was to check what happens to the waste of PURs and their blends with biopolymers after being in ordinary, random soil.

Samples (with a thickness of 1.0 ± 0.01 mm) were cut from the tested films in triplicate for each collection and then placed in a vacuum extractor with a drying bed for 6 h at 40 °C. The mass of individual samples was determined on an analytical balance. Plastic containers were prepared for each week of incubation, filled with soil to the height 3/4 of the container and the samples were placed to a depth of about 1 cm (Appendix A). The soil burial degradation process was carried out in a BINDER Model KBF-720 climatic chamber (Tuttlingen, Germany) at a temperature of 23 °C and a relative humidity (RH%) of 50% for the entire duration of the study. The sampling stage was established at 4, 12 and 36 weeks after placing the samples in the soil. On average, every two weeks, the contents of the containers were moistened with 5 mL of water (cleanliness class 3) and, in order to maintain an environment suitable for the development of soil microorganisms, once a month with 5 mL of vermicompost (Biohumus, Agrecol Sp. z o.o, Wieruszów, Poland). Biohumus solution was prepared according to the manufacturer’s instructions. Each time, the samples were taken out from containers, washed in distilled water and wiped with KIMTECH dust-free wipes. The drying stage was carried out in the same way as before incubation. After drying the samples to a constant mass, the weight loss of the samples was calculated, and then they were subjected to ATR-FTIR, ^1^HNMR, GPC, DSC, SEM and OM analysis. Moreover, the change in the contact angle of the sample surface after incubation in the soil was determined.

Attenuated total reflectance Fourier transform infrared spectroscopy (ATR FTIR) was used to determine the characteristic groups of polyurethanes. FTIR spectra were recorded with an attenuated total reflection (ATR Smart Orbit Accessory, Thermo Fisher Scientific, Madison, WI, USA) mode on a Nicolet iS 10 spectrometer (Thermo Scientific, Madison, WI, USA) with a diamond cell. A resolution of 4 cm^−1^ and a scanning range from 600 to 4000 cm^−1^ were applied, and 16 scans were taken for each measurement.

Nuclear magnetic resonance ^1^H NMR spectroscopy of PURs samples before and after 36 weeks of incubation was recorded using a Bruker-Advance spectrometer (Bruker BioSpin GmbH, Rheinstetten, Germany) operating at 600 MHz with Bruker TOPSPIN 2.0 software using CDCl_3_ as the solvent and tetramethylsilane (TMS) as the internal standard. Spectra were obtained with 128 scans, an 11 μs pulse width and a 2.66 s acquisition time.

The average molecular weight and molecular weight dispersity (M_w_/M_n_) of the polymers were determined using gel permeation chromatography (GPC-MALLS) with a differential refractive index detector Dn-2010 RI (WGE Dr. Bures, Berlin, Germany) and a multiangle laser light scattering detector DAWN EOS (Wyatt Technologies, Santa Barbara, CA, USA). GPC was performed using the following set of columns: GRAM gel guard, GRAM 100 Å, GRAM 1000 Å and GRAM 3000 Å (Polymer Standard Service, Mainz, Germany). The measurement was made in DMF supplemented with 5 mmol/L LiBr at 45 °C with a nominal flow rate of 1 mL/min. The sample solution was filtered prior to injection using a syringe PTFE filter with a pore size of 0.22 µm (GVS, Bologna, Italy). The polystyrene standards (Polymer Laboratories, Church Stretton, UK) with narrow molecular mass distribution were used to generate a calibration curve. The results were evaluated using PSS Win GPC Unity software from Polymer Standard Service (Mainz, Germany). 

Differential scanning calorimetry (DSC) analysis was conducted with a Setaram thermal analyzer (Setaram, Caluire, France). Indium and lead were used for calibration. Specimens (with mass of about 8 mg) were sealed in aluminum pans and scanned from 20 to 200 °C, with a heating rate of 10 C/min. All experiments were carried out in a dry flow of N_2_.

A scanning electron microscope (SEM) was used to investigate the structure and morphology of sample surface before and after incubation. Sample surfaces were analyzed by a Quanta FEG 250 SEM (FEI, Hillsboro, Oregon, USA). The SEM was equipped with an ET secondary electron detector. A total of 10 nm of gold layer was spattered on the surface to avoid charging of the sample. The images were taken at the sample surface in multiple places to confirm sample uniformity. The acceleration voltage was kept at 10 kV during the measurements and was constant.

Optical microscopy (OM) micrographs were taken using a Nikon microscope ALPHAPHOT-2 YS2-H (Japan) connected with Delta Optical DLT-Cam PRO 6.3MP USB 3.0. The micrographs were collected under reflected and transmitted light.

Static water contact angle measurements were carried out using the DataPhysics OCA 15EC contact angle system (DataPhysics instruments GmbH, Filderstadt, Germany) at ambient temperature. A 2 mL distilled water drop was used for each measurement and a photo of the drop was taken after 0, 1 and 3 min from immersing the drop on the sample surface. At least five measurements were made for each different system.

## 3. Results and Discussion

Macroscopic observations indicate the gradual degradation of each type of material with the time of exposure to soil. A characteristic feature is yellowing and browning of the surfaces of the tested samples (Appendix A). There was also an indelible darkening in the micropores resulting from the degradation of the sample. This indicates the activity of microorganisms. The dark spots are supposed to be remnants of bacterial colonies. The longer the samples were incubated in the soil, the more intense these changes were. The greatest number of discolorations, as well as numerous microcracks visible on the surface, was found for the samples PUR 20/5 + St (Appendix A).

### 3.1. The Weight Loss

Enzymatic hydrolysis is a heterogeneous process. The relatively high molecular weight of the enzymes hinders their action in the bulk of the polymer, hence the changes occurring first on the sample surface [21]. First, enzymes are adsorbed on the PUR surface, then they hydrolyze ester bonds (which is the dominant degradation route), and next they hydrolyze urethane bonds more slowly by an order of magnitude [22]. Thus, enzymes of soil-borne microorganisms can attack the surface of the PUR and cut macrochains into shorter ones by hydrolytic attack, thus causing surface erosion. This process depends on many factors: the polymer itself (its chemical composition, molecular weight, hydrophilicity, amorphism, surface roughness, crosslinking, etc.), abiotic (amount of water, soil pH, temperature, oxygen access, etc.) and biotic factors (microorganisms). Only when the molecular weight of the polymer is reduced to an appropriate size, microorganisms can assimilate these compounds and use them as a carbon source. The polycaprolactone segments are considered to be one of the slowest degrading polymers among all those that are biodegraded [23]. As shown in previous studies, blending of PCL with R,S-PHB in the soft segment increased the susceptibility of PURs to degradation under hydrolytic and oxidative conditions [17]. Moreover, increasing the amount of R,S-PHB in the PUR structure slightly accelerated the degradation process of the samples subjected to soil burial test (Figure 1A,B). PCL is known as a hydrophobic and semi-crystalline polymer [21], so its partial replacement with amorphous R,S-PHB should facilitate the penetration of water into the sample mass. Moreover, R,S-PHB as a synthetic equivalent of the natural PHB has a chemical structure recognizable by microorganisms living in the soil. Thus, although the amount of water in the soil was significantly lower than in phosphate buffer, the reduction in mass of PUR 10/5 in the soil was slightly higher than when they were hydrolyzed [17]. The acceleration of the kinetics of polyesterurethane degradation under the influence of microorganisms compared to abiotic factors has already been observed many times [24,25]. As expected, blending the polyurethanes with PLA and St increased the susceptibility of the samples to degradation. Both modifiers with a chemical structure found in the natural environment are rapidly biodegradable [26,27,28]. Despite only a small amount of them being present in the samples, both PUR 10/5 and PUR 20/5 blends degraded faster in the soil than pristine PURs. While PLA blends decreased their mass at a similar level as in the case of hydrolysis (Figure 1A,B), the degradation of PUR 20/5 + St in the soil was slower (Figure 1B) [17]. It is clearly visible how important in starch degradation is the presence of water, of which there is much less in the soil.

### 3.2. ATR-FTIR

FTIR was used to study the structural changes occurring in PURs and their blends after degradation in soil. Figure 2A shows the spectra of ATR-FTIR PUR 20/5 and its PLA and St blends before and after 36 weeks of the sample’s exposure on the soil. The FTIR spectrum of pristine PUR 20/5, PUR 20/5 + PLA and PUR 20/5 + St showed the presence of characteristic bands typical for PUR: stretching –NH (around 3370.0 cm^−1^), as well as I, II and III amide bands, respectively, at around 1720 cm^−1^, 1523 cm^−1^ and 1240 cm^−1^. The band at 1720 cm^−1^ is a multiplier of superimposed bands resulting from absorption of C=O groups from esters building soft segments and from urethane [29]. The urethane C=O band has a lower wavenumber than 1720 cm^−1^ and is visible as a bulge on the right side of the band (Figure 2B). The alkyl group bands (CH_2_ and CH_3_) are at around 2933 and 2863 cm^−1^. The absence of a band indicating the presence of OH groups in the ATR-FTIR spectrum of the PUR 20/5+St sample may be due to the low amount of St in the composite or its presence deep inside the sample, not on the composite surface.

The blending of PURs with PLA disrupted the order of the chains. In the exemplary PUR 20/5 and PUR 20/5 + St spectra, the band at 1722 cm^−1^ (stretching of C=O in ester groups) indicates the presence of hydrogen-bonded ester groups. However, in the case of PUR 20/5 + PLA, there is a clear shift of these bands towards higher wavenumber values, indicating the presence of free carbonyl groups (Figure 2B). Increase in the number of C=O groups in the PUR 20/5 + PLA blend, resulting from the presence of ester groups in PLA, compared to PUR 20/5, meant that some C=O groups are not linked by hydrogen bonds. However, after 36 weeks of burying the samples in the soil, secondary hydrogen interactions were formed due to the water in the soil, which is evident by slight shifting of the band to lower wavenumber values (Figure 2B). In the blend spectra, it can also be observed that there is a slight shift in the maximum of the broad band corresponding to the N-H stretching vibration (Appendix A), as in the case of PUR 20/5 + St, when the band maximum shifted from 3365.2 cm^−1^ to 3357.0 cm^−1^ after 36 weeks of exposure on the soil. It proves the enhancement of the hydrogen interactions between the –NH group of the urethane and the carbonyl group. Moreover, in the fingerprint region of PUR 20/5 + PLA, some changes in the course of the spectrum can be seen after the sample’s exposure to soil (Figure 2C). However, if the characteristic bands of groups potentially susceptible to hydrolysis are analyzed, it is clearly visible that the structural changes of PURs and their blends after exposure on the soil are small (Appendix A) [30,31,32]. As indicated by Oprea et al., the lack of significant changes in intensity for the bands of the NH (around 3360 cm^−1^) and CO (1520 and 1720 cm^−1^) groups indicates low biodegradation in the urethane region [31]. It is well-known that ester bonds are relatively easy to hydrolyze and are susceptible to enzymatic (microbial) cleavage. On the other hand, urethane moieties are much less susceptible to biodegradation and, as Trhlíková et al. [8] write, it is not yet clear whether they undergo enzymatic hydrolysis when they are present in the high-molecular PUR structure, or whether earlier fragmentation of the PUR network into shorter oligomers is required. Pilch-Pitera and Wojturska found biodegradation of urethane groups of PUR based on PCL after 15 weeks under composting conditions (temperature 58 °C, continuous aeration of the system, inoculation with thermophilic microorganisms and essential nutrients) [33]. They stated that the change in the intensity of the bands at 1180 and 1095 cm^−^^1^, associated with stretching vibrations of the absorption system of the C–O–C (ester), indicates the biodegradation of ester parts of PURs materials after the action of biotic factors (in that case there were fungals; in our study, these were all biotic factors contained in the soil). The intensity of peaks at 1045 and 1195 cm^−1^ increased, while the peak around 1095 cm^−1^ decreased after 36 weeks in soil, which is especially visible in the case of PUR 20/5 + PLA (Figure 2C). This confirmed that ester groups on the surfaces of the investigated sample were degraded.

### 3.3. ^1^HNMR

Figure 3 shows the ^1^H NMR spectrum of PUR 20/5 + PLA in CDCl_3_, in which all proton signals belonging to PCL_triol_, PCL_diol_, R,S-PHB, H_12_MDI and PLA segments are confirmed [29,34]. The band corresponding to the proton of the urethane group is found at 7.28 ppm in the spectra (blue box in Figure 3). At the same time, on both sides of it, the presence of small bands corresponding to the hydrogen interactions between the proton of the urethane group and O=C from urethane (7.18 ppm), and also between the NH urethane and the O=C ester (7.45 ppm) was found [29]. The intensities of these bands for the PUR 20/5 and PUR 20/5 + PLA samples did not change after incubation in soil. The probable swelling of starch in PUR 20/5 + St under the influence of soil conditions increased the hydrogen-type interaction between polyurethane chains, which is visible by increasing the intensity of the bands at 7.09 and 7.42 ppm (Appendix A) and confirmed FTIR results.

The protons and the corresponding number markings on the ^1^H NMR spectra are summarized in Table 2.

### 3.4. Molecular Weight Changes

Microorganisms contained in the soil release enzymes into the environment. Enzymes absorbed on the surface of the material can depolymerize macromolecules. Only when the molecular weight of the material is properly reduced, the water-soluble intermediate degradation products are released. These low-molecular particles can be absorbed directly into the microorganism cells, where most biochemical processes take place, and can be used there in the metabolic process. The end products of the total biodegradation process are H_2_O, CO_2_, CH_4_ and biomass [35]. Table 3 summarizes the values of the molecular weight M_n_ and M_w_ as well as the molecular-weight dispersity (M_w_/M_n_) of PUR 20/5 and its blend. The molecular weights of all polyurethane materials are similar. Moreover, as can be seen, the value of M_n_ practically did not change after incubation in soil, only in the case of PUR 20/5 + PLA did it slightly decrease. These observations are consistent with only the slight structural changes observed in the FTIR spectra. This change is incomparably smaller than when this material was subjected to hydrolytic degradation [17]. In that research, the number of average molecular weight of PUR 20/5 + PLA after 36 weeks of incubation in the hydrolytic solution was only 5900 Da. Probably, if the incubation time is extended, the weight loss of the samples from that moment would be very large, because, as is known [36], at molecular weight below 6000 Da, the polymer chains dissolve in the buffer. Other investigations show that the weight must drop to 3000 Da [25]. Despite low M_n_ reduction, the M_w_ value and consequently the molecular-weight dispersity of the tested polymer materials changed. Sikorska et al. stated that when the bonds near the center of the polymer are more stable than those near the ends of the chain, then M_w_/M_n_ increases above the value of 2 [37]. For PUR 20/5 the increase in M_w_/M_n_ was observed. It indicates changes in the length of the polyurethane chains, connected with their cutting under the influence of soil conditions. This leads to shorter chains, but molecular weight is too high for these chains to be washed out of the sample, resulting in an increase in the dispersity of the polyurethane. On the other hand, a reduction in dispersity index may indicate elution of short polymer chains, as in the case of PUR 20/5 + PLA and PUR 20/5 + St. The addition of both biopolymers to the PUR 20/5 matrix probably resulted in the loosening of the chain network and easier elution of low-molecular weight fractions during incubation in the soil, which resulted in the reduction of molecular-weight dispersity. However, the visible weight loss of the samples (Figure 1A,B) probably results from the washing out of the inclusions from PLA and St, and not from the significant hydrolysis of the PUR matrix itself. This indicates no change in molecular weight, as well as changes on the surface of the samples observed on microscopic photos (Table 4, Appendix A).

### 3.5. Changes of Thermal Properties

The tested polyurethanes and their blends have low melting points of the crystalline phase of soft segments (Table 5). Instead of the expected increase in crystallinity after degradation, a slight decrease was found. The melting enthalpies of these segments were slightly changed after sample’s exposure to soil, which indicates only slight structural changes in the polymers. This confirms the observations of the ATR-FTIR and ^1^HNMR spectra. Slightly lower enthalpies of PUR 20/5 and its blend, indicating lower crystallinity of these samples, may be the reason for higher degradation kinetics and higher mass loss (Figure 2 and Figure 3) than in the case of PUR 10/5 and PUR 10/5 + PLA [25].

### 3.6. Surface Morphology Changes

The surface of pristine PURs is similar. Regular bulges can be seen on the PUR 10/5 and PUR 20/5 surfaces. They are probably the results of the presence of certain crystalline forms. There are more bulges on the PUR 10/5 surface, which results from its slightly higher crystallinity (ΔH_PUR10/5_ > ΔH_PUR20/5_) (Table 5). This can be better seen in the images taken from the OM (Appendix A) than in those taken from SEM (Table 4). The influence of time of PURs and the blend’s exposure to soil on their biodegradability can very well be seen in the SEM images. After burying in the soil, small cracks and holes (red arrows in Table 4) appeared on the surface. The formation of such pits and holes on the surface of polyesterurethanes, after the action of particular strains of microorganisms, has often been observed [38]. As Podzorova and Tertyshnaya point out, despite a smaller reduction in the mass of the PLA/PE blend after exposure to soil in the field than in the laboratory, the changes in the surface of the samples from the field were greater [39]. This was related to the activity of microorganisms, the amount of which in laboratory conditions gradually decreased because they were not replenished. However, there was the constant (about 22 °C) temperature and high humidity (about 60%) that caused the higher degree of degradation of these samples, which of course increased with the increase in the amount of PLA.

As is shown in Table 4, Appendix A, much larger changes were noticed on the surface of the modified PURs. Since the PURs and their blends were partially transparent, transmitted-light OM images were also taken (Appendix A). The OM images of the blends show clear darkness in the areas of inclusions made of biodegradable PLA and starch. This may indicate the presence of microbial colonies as well as pitting on the surface associated with the degradation of these modifying polymers.

Both modifiers are biodegradable, despite PLA being a hydrophobic material [39]. In contrast, starch is a highly hydrophilic material, like guar gum, that draws moisture into the blend, allowing the growth of starch-processing microorganisms. It was known that water with microorganisms penetrated into the matrix of polyetherurethane acrylates and their guar gum composites and destroyed the integrity of the polyurethane film [40]. Indeed, in the transmitted-light OM images (Appendix A) of the PUR 20/5 + St samples, there are clear blackouts in the polyurethane matrix indicating the presence of microbial colonies in the blend bulk. The SEM images show the degradation of PLA round inclusions in the matrix of PUR 10/5 + PLA and PUR 20/5 + PLA (Table 4). The observed changes on the surface of the samples indicate that the enzymatic degradation process has started. Extending the time of incubation in the soil makes for easier penetration of water through the damaged surface of the samples and accelerates the kinetics of the degradation process.

### 3.7. Contact Angle Changes

Changes in the contact angle can be used to assess the influence of the degradation environment on the morphology of the surface layer of the material and the progress of the degradation process [41]. It is obvious that the hydrophilicity and the associated contact angle of the polymer surface depend on the structure of the polymer itself (stereochemistry, the number of adjacent methyl groups, the presence of nucleophiles, etc.), but also on the roughness of the surface [42]. As can be seen in Figure 4 and Figure 5, the contact angle of the surfaces of all samples gradually decreased over time (detailed data are given in Appendix A). Due to their segmented structure, PURs have the ability to maintain a hydrophilic–hydrophobic balance [43]. Upon contact with a drop of water, hydrophilic parts of the chains (such as R,S-PHB) moved to the surface of the sample, reducing the contact angle [44]. However, in the case of composites with PLA, a significant increase in surface roughness after sample degradation in soil increases the contact angle. This increase is especially marked with PUR 20/5 + PLA. While examining the biodegradation in compost and in water of biobased PURs, it was found that their surface structure is susceptible to enzymatic hydrolysis, as a result of which interfacial interactions and thus the value of surface free energy are subject to changes [45]. The increase in energy is associated with an increase in the hydrophilicity of the surface due to the formation of new polar groups as a result of the hydrolysis reaction. The amount of change in this energy can be influenced by both the chemical heterogeneity of the surface and its roughness, causing changes at the interface between the measuring liquid/test surface/air. It was noted that there was a 15% decrease in the contact angle of PUR based on H_12_MDI after 45 days of degradation in the compost [45].

## 4. Conclusions

Aliphatic, branched polyesterurethanes based on R,S-PHB and additionally modified with PLA and St were obtained and exposed to the soil. The study of susceptibility to degradation was carried out by burying the samples in the anthropological soil in conditions simulating the average moisture and temperature in Poland in the months from April to October. This experiment allows us to predict the behavior of the investigated polymer samples after their disposal in landfills. However, as Podzorova and Tertyshnaya point out, placing the samples in the ground under natural conditions meant that they are additionally exposed to temperature changes, including freezing and high temperatures up to 40–50 °C, as well as periodic flooding with water [39]. This is the reason for increased mechanical and chemical degradation. These processes, acting synergistically with enzymatic degradation, would significantly accelerate the degradation of polymers.

Burying the samples in the soil for 36 weeks resulted in the loss mass of the samples (up to 7.9 wt.%) and changed their surface morphology. However, only slight changes were found: in the molecular weight and dispersity of these polymers, in the chemical structure and in thermal properties. The molecular weight practically did not reduce, while the M_w_/M_n_ ratio changed, which indicates: (i) chain cutting to shorter molecular weights—in the case of the pristine PUR; (ii) elution of shorter chains—in the case of blends with PLA and St. The FTIR and ^1^HNMR spectra indicate only slight hydrolysis of the ester groups and interference with the interactions of the urethane bonds. It can be seen, however, that at this stage of the investigation, the presence of small PLA and St inclusions that have been washed out of the PUR matrix has a greater influence than the presence of R,S-PHB. Probably the extension of the time of exposure to the soil of the samples would cause the hydrolysis of amorphous R,S-PHB and thus also the degradation of PUR. The results of this study show that blending the branched PUR based on R,S-PHB with only a small amount of degradable PLA and St allows slow but gradual (bio)degradation of these materials when they enter a municipal or wild landfill, where conditions for the degradation of polymers are much worse than in industrial compost.

## Data Availability

Data sharing not applicable.

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
