# Peer review of "Susceptibility to Degradation in Soil of Branched Polyesterurethane Blends with Polylactide and Starch"

_polymers, 2022, doi:10.3390/polym14102086_

Round 1

Reviewer 1 Report

The paper presents the results of testing the susceptibility to degradation in soil of branched polyesterure- thane (PUR) based on poly([R,S]-3-hydroxybutyrate) (R,S-PHB), modified with poly([D,L]-lactide) (PLA) and starch (St). The results indicate that these materials are subject to enzymatic degradation caused by the presence of microorganisms in the soil. Chemical modification of branched polyesterourethanes with R,S-PHB and their physical blending with small amount of PLA and St resulted in a slow but progressive degradation of the samples. But the study didn't go far enough and the research is not enough novelty, so the paper is not recommended for publication in polymers.

Author Response

The paper presents the results of testing the susceptibility to degradation in soil of branched polyesterurethane (PUR) based on poly([R,S]-3-hydroxybutyrate) (R,S-PHB), modified with poly([D,L]-lactide) (PLA) and starch (St). The results indicate that these materials are subject to enzymatic degradation caused by the presence of microorganisms in the soil. Chemical modification of branched polyesterourethanes with R,S-PHB and their physical blending with small amount of PLA and St resulted in a slow but progressive degradation of the samples. But the study didn't go far enough and the research is not enough novelty, so the paper is not recommended for publication in polymers.

Reply to review:

We would like to thank the Reviewer for reading and reviewing our manuscript.

We agree that a lot of research has been carried out on the susceptibility of polyesterurethanes in soil. Also the degradability of the polyurethanes modified with biopolymers and its synthetic counterparts has already been investigated.

However, the need to investigate these specific polyurethanes (obtained from R,S-PHB and PCL as well as cycloaliphatic H12MDI), and how to influence the susceptibility of these materials in various environments, arose from discussions with an industry representative. This manager is looking for a polyurethane that would be synthesized from non-toxic substrates and, moreover, would slowly degrade after its end of use. We have already tested these polymers in a hydrolytic and oxidizing environment (Polymers 13, 1202; https://doi.org/10.3390/polym13081202), hence the assumed comparison time of 36 weeks of incubation in soil. The investigation is also underway in the natural environment of seawater, and incubation is planned in an activated sludge at a waste treatment plant. While browsing the available library resources, we did not find a monograph describing the results of the study of degradation in soil of polyurethanes with similar chemical composition. In the library databases it can be find our earlier papers on these materials, but not on their degradation in soil.

Therefore, the studies described in the manuscript are preliminary studies to show whether it is worth starting work on these polyurethanes and on ways to change their degradation kinetic in the natural environment. The intention of this work was to investigate whether these polyurethanes, under the influence of biopolymer modification, will degrade faster after the end of their service life. Unfortunately, in Poland we have a big problem with polyurethane waste deposited in wild and municipal landfills. The conditions in these places are different; they cannot be averaged to determine if they are optimal for the biodegradation of these materials. We will also never have a guarantee that the environment will contain just microorganisms capable of assimilating such chemical compounds. Therefore, it seemed important to us to see whether in ordinary soil, where such polymer waste may happen by chance, polyurethane made of degradable substrates and additionally modified with biopolymers, will actually biodegrade over time.

We hope the Reviewer will understand our intentions and recognize the need to show these preliminary results.

Reviewer 2 Report

The manuscript of Brzeska et al is devoted to the study of the possibility of reducing polymeric pollutants in natural conditions after polymer modification.

the authors have conducted a long study on the transformation of the modified polyesterurethane using good methods of analysis. As a result of a long-term experiment, a slight change in the structure was shown, which the authors attributed to the activity of microorganisms.

If the role of microorganisms in this process is discussed, then, in my opinion, the authors lacked two controls. The first is the conduct of the experiment in a sterile soil to differentiate the role of microorganisms and soil factors. Secondly, the authors could well carry out similar studies under the action of the solar spectrum. In this version of the experiments, the authors simply confirmed the fact that the decomposition of polymers in nature is a long-term matter.

From minor remarks.

Line 17, abstract. The first sentence should be rephrased, as it is unlikely that anyone would make polymer waste biodegradable. Probably, we are talking about using polymers that can be subject to microbial degradation.

Line 121 - what kind of anthropogenic soil was used? It is necessary to give at least some definition of it, the presence of organic material, the number of cultivated microorganisms, microbial profiling, GPS coordinates of the sampling point. From the point of view of microbiology, the setting of the experiment requires a more detailed description.

lines 159-161. Check if all words are spelled separately

In terms of chemical analysis, the authors presented significant results. From the point of view of microbial degradation of synthesized polymers - poor work.

Author Response

We are very grateful to the reviewer for his effort to review our manuscript and for any suggestions.

Here are our responses to the reviewer's comments and suggestions. Changes in the manuscript are marked in yellow.

The manuscript of Brzeska et al is devoted to the study of the possibility of reducing polymeric pollutants in natural conditions after polymer modification.

the authors have conducted a long study on the transformation of the modified polyesterurethane using good methods of analysis. As a result of a long-term experiment, a slight change in the structure was shown, which the authors attributed to the activity of microorganisms.

If the role of microorganisms in this process is discussed, then, in my opinion, the authors lacked two controls. The first is the conduct of the experiment in a sterile soil to differentiate the role of microorganisms and soil factors. Secondly, the authors could well carry out similar studies under the action of the solar spectrum. In this version of the experiments, the authors simply confirmed the fact that the decomposition of polymers in nature is a long-term matter.

We agree with the Reviewer's remark that this research would perfectly complement the obtained results. The tested materials are polyesterurethanes, additionally modified with biopolymers, therefore the main mechanism of their degradation is hydrolysis. The susceptibility to this reaction was previously checked in phosphate buffer under sterile conditions (NaN3 was added to the buffer solution). This degradation has proven slow to progress. Hence, we decided that it was pointless to study the degradability of these materials in soil containing less water than in a buffer solution, in which there would also be no microorganisms. Therefore, we decided not to test in sterile soil. However, we set up test in an ordinary garden soil under natural conditions (for the assumed period of 36 weeks), for comparison. Unfortunately, the organisms living in the garden (probably voles) dispersed the samples. We were able to collect only some samples after 4 weeks of incubation (photos of the saved samples are provided below). The lack of changes in the mass and surface of these samples indicates an important influence of the temperature, which was constant and quite high (23 °C) in laboratory conditions, on the degradation process of these polymers, as well as the influence of the amount of moisture in the degradation environment, which is obvious. However, since we only have some samples from the first collection, we cannot put them into the work. Unfortunately, we also have no technical means to test the susceptibility of these samples to degradation under the influence of the solar spectrum.

The sample images - in attached material.

From minor remarks.

Line 17, abstract. The first sentence should be rephrased, as it is unlikely that anyone would make polymer waste biodegradable. Probably, we are talking about using polymers that can be subject to microbial degradation.

Of course, we agree with the Reviewer that this sentence is too far-fetched an assumption. It was corrected in the Manuscript to:

“The very important method of reducing the amount of polymer waste in the environment is the introduction to the market of polymers susceptible to degradation under the influence of environmental factors.”

Line 121 - what kind of anthropogenic soil was used? It is necessary to give at least some definition of it, the presence of organic material, the number of cultivated microorganisms, microbial profiling, GPS coordinates of the sampling point. From the point of view of microbiology, the setting of the experiment requires a more detailed description.

The horticultural soil, used for the study, was not cultivated, therefore not fertilized with natural or synthetic fertilizers. It was taken from a garden in the northern part of Poland (GPS coordinates: 54.528368455536864, 18.289577253812595), with a warm, temperate transitional climate Soil sieve analysis is given in Figure S1, and density and elemental composition before and after the addition of vermicompost are summarized in Table S1.

Table S2 lists the total numbers of bacterial and fungal colonies, with some bacteria in detail. Unfortunately, we are no able to reproduce now how the number of colonies changed during the course of the experiment. This would mean setting up the entire study all over again. On the other hand, the Reviewer's suggestions prompted us to check whether the presence of polyurethane samples in the soil influences the development of microorganisms in the soil. There were no such studies, because the intention of this study was to investigate whether these polyurethanes, under the influence of biopolymer modification, would degrade faster after the end of their service life. So, in fact, we were interested in the changes taking place in the material itself, and not in the environmental factors that these changes will cause. Unfortunately, in Poland we have a big problem with polyurethane waste deposited in wild and municipal landfills. The conditions in these places are different; we will never be able to average them to determine if they are optimal for the biodegradation of these materials. We will also never have a guarantee that the environment will contain just microorganisms capable of assimilating such chemical compounds. Therefore, it seemed important to us to see whether in ordinary soil, where such polymer waste may happen by chance, polyurethane made of degradable substrates and additionally modified with biopolymers, will actually biodegrade over time.

lines 159-161. Check if all words are spelled separately

We checked the text in lines 159-161 of the original manuscript and in our opinion the notation is correct. However, if the Reviewer thinks something is wrong, please give us a hint.

Reviewer 3 Report

This manuscript entitled “Susceptibility to Degradation in Soil of Branched Polyesterurethane Blends with Polylactide and Starch” by J. Brzeska et al reported the biodegradation analysis of branched polyesterurethane (PUR) based on poly([R,S]-3-hydroxybutyrate) (R,S-PHB), modified with poly([D,L]-lactide) (PLA) and starch (St). The results and discussion on the biodegradation of plastic films are plausible. The reviewer recommends the revisions before the publication.

  1. Please add the scheme drawing the synthetic preparation procedures of branched polyesterurethane (PUR) for the smooth understanding of polymer structure.

- The term “poly(ester-urethane) might be more accurate than “polyesterurethane”.

Typo-error: polyesterourethanes (abstract)

  1. To understand the biodegradation test results in this study, it is very important to analyze the fundamental soil characters. This is because if the soil conditions varied, the biodegradation results will also be changed. It needs more information on soil (NOT restricted to L121-125). Please refer to the references below: Table S5 in Green Chem. 2021, 23, 6953. Water, soil organic matter, nitrogen contents, bulk density, and etc will be useful.

  1. How the authors prepared the testing specimen films?

Author Response

We are very grateful to the reviewer for his effort to review our manuscript and for any suggestions.

Here are our responses to the reviewer's comments and suggestions. Changes in the manuscript are marked in yellow.

This manuscript entitled “Susceptibility to Degradation in Soil of Branched Polyesterurethane Blends with Polylactide and Starch” by J. Brzeska et al reported the biodegradation analysis of branched polyesterurethane (PUR) based on poly([R,S]-3-hydroxybutyrate) (R,S-PHB), modified with poly([D,L]-lactide) (PLA) and starch (St). The results and discussion on the biodegradation of plastic films are plausible. The reviewer recommends the revisions before the publication.

  1. Please add the scheme drawing the synthetic preparation procedures of branched polyesterurethane (PUR) for the smooth understanding of polymer structure.

Manuscript was supplemented with a scheme for the synthesis of modified polyesterurethanes (Scheme1), as well as a schematic representation of the chemical structure of polyesterurethanes (Scheme2) and their blends with biopolymers (Scheme3) (page 3).

- The term “poly(ester-urethane) might be more accurate than “polyesterurethane”.

Typo-error: polyesterourethanes (abstract)

When quoting the name "polyesterurethane", we followed Compendium of Polymer Terminology and Nomenclature IUPAC Recommendations 2008, as well as publications published in magazines of such publishing houses as Springer, Elsevier, Willey, De Gruyter, etc., in which polymers are written in this way. But we understand that perhaps it would be clearer to call them "poly(ester-urethane)". If the Reviewer wishes us to change the name, we will of course do so.

  1. To understand the biodegradation test results in this study, it is very important to analyze the fundamental soil characters. This is because if the soil conditions varied, the biodegradation results will also be changed. It needs more information on soil (NOT restricted to L121-125). Please refer to the references below: Table S5 in Green Chem. 2021, 23, 6953. Water, soil organic matter, nitrogen contents, bulk density, and etc will be useful.

The horticultural soil that was used for the study, was not cultivated, therefore not fertilized with natural or synthetic fertilizers. It was taken from a garden in the northern part of Poland (GPS coordinates: 54.528368455536864, 18.289577253812595), with a warm, temperate transitional climate. The containers with soil and polymer samples were placed in a chamber with 50% relative humidity and additionally wetted with 5 ml of water every 2 weeks (purity class 3). For this reason, the water content varied slightly during the test and an average value cannot be given. Especially that once a month, an aqueous solution of vermicompost was also added. Soil sieve analysis is given in Figure S1, and bulk density and elemental composition before and after the addition of vermicompost are summarized in Table S1. Table S2 lists the total numbers of bacterial and fungal colonies, with some bacteria in detail.

It is obvious that the conditions in the environment to which the waste ends up after use may be radically different. Of course, it is important to determine the kinetics of sample degradation due to the influence of specific microorganisms or enzymes. However, you never know to what place (whether it is a municipal or wild landfill) this polymer waste will be dumped. And it may turn out that microorganisms capable of bioassimilating this material do not exist in this environment, or there is a negligible amount of them. The essence of our study was to determine whether in such random soil any changes in the polymer samples took place, which would indicate that these materials could potentially gradually degrade after the end of their use. This would be one of the ways to solve the problem of polyurethane waste, which we observe in the environment in our country.

  1. How the authors prepared the testing specimen films?

In the manuscript, in the section Methods, the method of forming polyurethane films is described (Figure 1). As it was shown in the manuscript, the samples (with a thickness of 1.0± 0.01 mm ) for degradation measurements were cut from the tested films in triplicate for each collection, and then placed in a vacuum extractor with a drying bed, for 6 hours at 40 °C.

As all three samples of a given polyurethane material were placed in one container with soil, they were cut into three different shapes (about 1 cm2) in order to prevent their signing and thus not to affect the parameters (e.g. pH) of the soil. After the incubation time, the samples were washed, dried to constant weight and weighed, and then their chemical structure, molecular weight, thermal properties and surface morphology were determined.

Round 2

Reviewer 1 Report

The manuscript has been revised carefully, it can be published in the journal after polish the english expression.

Reviewer 2 Report

The authors have made some changes to the text of the manuscript. I am glad that my comments gave an idea for further research. Good luck to the authors.

Reviewer 3 Report

The authors addressed all issues properly, therefore the reviewer recommends the publication.